# Vitamin D and Calcium Milk Fortification in Pregnant Women with Periodontitis: A Feasibility Trial

**DOI:** 10.3390/ijerph17218023

**Published:** 2020-10-30

**Authors:** Amanda Rodrigues Amorim Adegboye, Danilo Dias Santana, Paula Guedes Cocate, Camila Benaim, Pedro Paulo Teixeira dos Santos, Berit Lilienthal Heitmann, Maria Cláudia da Veiga Soares Carvalho, Michael Maia Schlüssel, Maria Beatriz Trindade de Castro, Gilberto Kac

**Affiliations:** 1School of Human Sciences, Faculty of Education, Health and Human Sciences, University of Greenwich, Park Row, London SE10 9 LS, UK; 2Nutritional Epidemiology Observatory, Department of Social and Applied Nutrition, Institute of Nutrition Josué de Castro, Federal University of Rio de Janeiro, 21941-902 Rio de Janeiro, Brazil; dias.danilo@hotmail.com (D.D.S.); camilabenaimnutri@gmail.com (C.B.); dr2p@hotmail.com (P.P.T.d.S.); mariaclaudiaveigasoares@yahoo.com.br (M.C.d.V.S.C.); mbtcastro@gmail.com (M.B.T.d.C.); gilberto.kac@gmail.com (G.K.); 3Department of Bioscience and Physical Activity, School of Physical Education and Sports, Federal University of Rio de Janeiro, 21941-599 Rio de Janeiro, Brazil; paulacocate@gmail.com; 4Research Unit for Dietary Studies at the Parker Institute, Bispebjerg and Frederiksberg Hospital, The Capital Region, Denmark and Section for General Medicine, Institute of Public Health, Copenhagen University, 2000 Copenhagen, Denmark; Berit.Lilienthal.Heitmann@regionh.dk; 5The EQUATOR Network–UK Centre, Centre for Statistics in Medicine, Nuffield Department of Orthopaedics, Rheumatology and Musculoskeletal Sciences, University of Oxford, Old Road, Oxford OX3 7LD, UK; michael.schlussel@csm.ox.ac.uk

**Keywords:** feasibility, acceptability, adherence, attrition rate, periodontal therapy, milk fortification, pregnancy

## Abstract

This study aims to assess the acceptability, adherence, and retention of a feasibility trial on milk fortification with calcium and vitamin D (Ca + VitD) and periodontal therapy (PT) among low income Brazilian pregnant women with periodontitis (IMPROVE trial). This 2 × 2 factorial feasibility trial used a mixed-methods evaluation. In total, 69 pregnant women were randomly allocated to four groups: 1. fortified sachet with Ca+VitD and milk plus early PT (throughout gestation); 2. placebo and milk plus early PT; 3. fortified sachet with Ca+VitD and milk plus late PT after childbirth; 4. placebo and milk plus late PT. Data were collected via questionnaires, field notes, participant flow logs, treatment diary, and focal group discussions. Quantitative and qualitative data were analysed using appropriate descriptive statistics and content analysis, respectively. Eligibility rate (12%) was below the target of 15%, but participation (76.1%) and recruitment rate (2 women/week) exceeded the targets. Retention rate (78.6%) was slightly below the target (80%). Adherence to the PT was significantly higher in the early treatment groups (98.8%) compared to the late treatment groups (29%). All women accepted the random allocation, and baseline groups were balanced. There was no report of adverse events. This multi-component intervention is acceptable, well-tolerated, and feasible among low-risk pregnant women in Brazil.

## 1. Introduction

Periodontitis, a gingival bacterial infection causing a breakdown of tooth-supporting structures, is a common condition in women of reproductive age [1]. Due to hormonal changes during gestation, pregnant women are prone to develop periodontitis or worsening existing gingival inflammation [2]. Evidence shows that periodontitis can influence gestational outcomes, maternal systemic health, and overall wellbeing [3,4,5]. Systematic reviews with meta-analysis have consistently reported that periodontitis increases the risk of premature birth, low birth weight [6,7], and pre-eclampsia [8]. However, there are still conflicting results regarding the increased risk of gestational diabetes [9].

A recent meta-analysis including four randomised controlled trials (RCT) found that nonsurgical periodontal therapy (PT) during pregnancy compared with an untreated group among women with chronic periodontitis did not decrease the risk of adverse pregnancy outcomes and maternal and neonatal inflammatory biomarkers [10]. It can be argued that the PT was not sufficiently intense or delivered early enough to prevent disease progression [11]. However, detailed information regarding intensity, fidelity, and adherence to the interventions was generally not provided. These interventions were performed in different settings and included diverse populations making it difficult to evaluate why effectiveness was limited [10]. Additionally, the meta-analysis included RCTs applying conventional nonsurgical periodontal treatment as the sole intervention, [10] and there is evidence suggesting that supplementation of vitamins and minerals, particularly vitamin D and calcium, might prevent the development of or delay the progression of periodontitis [12]. Therefore, further well-designed, long-term RCTs are still needed to evaluate the potential clinical benefit of vitamin D, and calcium supplementation as a co-adjunct treatment for periodontitis during pregnancy.

Although RCT is considered the most rigorous type of study design for evaluating the efficacy of interventions [13], low acceptability, adherence to treatment regimens, and retention rate can impact the potential effectiveness of interventions. A critical evaluation of these factors is crucial in informing future development and delivery of RCTs.

The IMPROVE feasibility trial was designed to assess the feasibility of a multi-component intervention including PT and consumption of fortified milk with calcium and vitamin D to improve the metabolic and inflammatory profile of pregnant women with periodontitis. In this paper, we evaluated the feasibility of the IMPROVE trial to inform the design of a large-scale and definitive RCT of effectiveness. Thus, this study aims to evaluate the operational aspects of the study design (e.g., random allocation), data collection, participation rate, retention of participants, acceptability, and tolerability, and describe factors associated with adherence to the intervention. Evaluation of the recruitment strategy such as barriers and facilitators to recruitment has been reported elsewhere [14].

## 2. Methods

### 2.1. Study Design, Randomisation Procedures, and Ethics

This is a 2 × 2 factorial randomised feasibility trial with a parallel process evaluation. The detailed study protocol has been described elsewhere [15]. The study applied a concealed randomisation, using permuted block sizes to ensure that groups were balanced periodically and stratified by smoking status. The randomisation was performed remotely via an online system developed by Sealed Envelope Ltd. (London, UK).

An explanation of study procedures was given verbally to all pregnant women invited to the trial and provided in the patient information sheet. Study enrolment occurred after receipt of informed written consent. This trial was registered in the clinicaltrials.gov database (NCT03148483) on 11 May 2017 and approved by the Ethics Committee of Maternity School of the Federal University of Rio de Janeiro-Brazil (approval reference number 1.516.656).

### 2.2. Setting

Participants were recruited from a public health care centre located in a low socioeconomic area in Duque de Caxias in Rio de Janeiro state in Brazil. The health care centre offers free prenatal care for low-risk pregnant women living within the catchment area.

### 2.3. Eligibility

Adult (18 years or older) women with a low-risk pregnancy, up to 20 weeks gestation at 1st prenatal visit, cognitively and physically able to complete an interview and oral examination, diagnosed with periodontitis and willing to participate (including the provision of blood samples) were considered eligible for the trial. Women with a diagnosis of HIV/AIDS, psychosis, diabetes before gestation, thyroid disease, disorders causing vitamin D hypersensitivity (e.g., sarcoidosis and other lymphomatous disorders), lactose intolerance and/or milk allergy, history of renal stones, family history of renal stone and hyperparathyroidism, extensive dental cavity and decay, use of braces, intake of antibiotics or any immune-suppressants or medication which affects vitamin D/calcium metabolism, consumption of ≥4 servings/day of dairy products, or taking vitamin D supplements at >400 IU/day were considered not eligible for the feasibility trial.

Women were invited to participate at their 1st prenatal visit by a nurse and reviewed against the inclusion/exclusion criteria (fully described elsewhere [15]), and then underwent a dental screening performed by a single trained dentist. Those who screened positive for the presence of periodontitis were invited to enter the study and undergo a full periodontal examination at baseline (T0) and after childbirth (T2). The presence of periodontitis was defined as ≥1 tooth with at least one site with ≥4 mm of clinical attachment loss (CAL) and the presence of bleeding on probing (BOP) on the same site.

Participant flow throughout the study is outlined in a CONSORT diagram (Figure 1).

### 2.4. Study Intervention Groups and Blinding

The study included four intervention groups without cross-over:Group 1—consumption of one sachet with powdered milk fortified with calcium and vitamin D twice a day and PT throughout gestation (early therapy).Group 2—consumption of a placebo sachet with powdered milk (plain milk) twice a day and PT throughout gestation.Group 3—consumption of one sachet with powdered milk fortified with calcium and vitamin D twice a day and PT after childbirth (late therapy).Group 4—consumption of a placebo sachet with powdered milk twice a day and PT after childbirth.

Women were asked to reconstitute 20 g of powder semi-skimmed milk and 2 g of fortification sachet containing calcium (CAPOLAC 500 mg) and vitamin D_3_ (500 IU)) in 200 mL potable water for each serving. They were also asked to consume the milk alone or blended in other preparations (i.e., kneaded fruits, fruit smoothies, yogurt, or porridge) during breakfast or afternoon snack to avoid concomitant intake of prenatal iron supplements routinely prescribed for consumption during main meals (lunch and dinner).

Periodontal examination was performed in the full mouth at six sites per tooth using North Carolina periodontal probes, a dental mirror and gauze, but without X-rays. Oral examination and treatment procedures were performed by two independent, calibrated, and trained dentists. The dentists always calibrate their probing force using a scale before the clinical examination. The recommended probing force was approximately 20 g of pressure. Although participants were blinded to the fortification allocation, due to the nature of the dental intervention, full blinding was not possible as participants knew whether they had been allocated to early or late PT. However, the dentist who performed the outcome assessment was blinded to the intervention allocation.

### 2.5. Outcomes and Data Sources

Several operationalised definitions of feasibility, acceptability, adherence, tolerability, and retention, with a priori specified threshold criteria were used in this study to assist in the pragmatic judgment on whether to accept, modify, or reject study components.

Feasibility: this domain included suitability of the study design, random allocation into intervention groups, and operational aspects of data collection procedures. This was defined as the extent to which participants considered the study design and data collection appropriate. Feasibility was assessed via participation rate, field notes on data collection procedures and qualitative data from monthly visits, and focus group discussions on the adequacy of study design. An online end-of-study evaluation survey, which included closed and open-ended questions regarding data collection experience, was completed by five members of the research team involved in data collection and fieldwork.

Recruitment: Patient, study protocol-, and setting-related factors associated with women’s ineligibility and refusal to participate in the study have been fully described in a previous publication [14]. In this paper, only quantitative indicators regarding participation, eligibility, and recruitment rates are presented.

Acceptability of the intervention: defined as the extent to which participants considered the intervention including the consumption of fortified milk and PT appropriate or whether they liked it. A five-point Likert-scale question regarding the acceptability of the milk powder was answered by 62 women in T0 (up to 22nd-week gestation) and 55 women in T1 (between 30 and 38 weeks of gestation). The response options were disliked very much, dislike, neither like nor dislike, like, and like very much. Acceptability was assessed via qualitative data on views of dental care treatment and consumption of milk and barriers and facilitators to the intervention collected via focus group discussions, feedback notes, follow-up visits, and calls.

Adherence to the intervention: defined as the degree to which the behaviour of participants corresponded to the intervention assigned to them and the level of compliance with the intervention protocol. At the end of each month, the pregnant women reported the number of sachets not consumed back to the researchers. Information on the number of sachets provided and consumed was recorded in a log-file for each participant. Adherence was calculated by the proportion of the self-reported number of sachets consumed out of the total number of sachets offered to participants. The value can vary between 0 and 1, and the closer the value is to 1, the greater the adherence to the fortified or plain sachet. Likewise, the adherence to the PT was calculated as the proportion of the number of therapy sessions completed out of the total number of therapy sessions offered. Each woman was entitled to receive up to five PT sessions as necessary.

Tolerability of the intervention: defined by patients’ ability to endure the intervention without experiencing complications or harm. This was measured by the number of serious adverse events (SAEs) notified during the study and participant complaint on feeling burdened or frustrated with data collection or by taking part in the study.

Retention: defined as the proportion of women who did not discontinue participation. The numbers of dropouts in each study group and follow-up points were also calculated. Completeness of outcome assessment was measured by the proportion of participants who provided full data on clinical outcomes at baseline, throughout pregnancy, and up to 6–8 weeks postpartum.

## 3. Parallel Process Evaluation

A process evaluation framework was developed to assist the content analysis by generating themes related to four main categories: (1) dietetics and culinary skills; (2) sharing of food with other family members; (3) health care needs, dealing with pain, access to health care centre; and (4) social support network and social challenges in life. A matrix was created before the data collection with the four main categories, which were subdivided into two levels: favourable and unfavourable factors and events. The categories were informed by preliminary focus group discussions performed with women with similar socioeconomic status of trial participants prior to the trial commencement. These women did not take part in the trial. Details about preliminary focus groups are provided elsewhere [14,15]. This matrix was created to facilitate data collection and systematically organise the themes and quotes from participants. The research team was trained to continually fill in the framework of content previously structured with data from the pilot focus group.

All women actively enrolled in the study at the time were invited to participate in the focus group followed by a social event held at the health care centre. The first focus group was held at the beginning of the study with women enrolled in the study for at least one month and the second one was performed with women in the third trimester (T1) or after childbirth (T2). Twenty-six women were invited for the first focal group, and 13 took part, while 54 women were invited for the second focus group and 10 took part. In total, 23 enrolled participants took part in two focus groups. Additionally, one-to-one visits with participants were conducted throughout the study to complement focus group data. In each monthly face-to-face visit, women were asked about the occurrence of any adverse events, barriers to adherence, and satisfaction with the intervention. Phone calls were made to those who did not attend the visits. Sentences and phrases reported by the women were recorded and added to the matrix.

## 4. Analysis

To evaluate the feasibility of the IMPROVE trial, an adapted checklist based on guidelines for reporting feasibility trails by Thabane et al. [16] was used. This checklist systematically describes the decision-making criteria on whether to (1) accept the original components of the current study protocol, (2) modify them, or (3) reject them (Table 1). A similar tailored checklist has been applied in a previous feasibility trial of a non-pharmaceutical intervention [17] to provide insights regarding the interpretation of an a priori threshold for the feasibility criteria on different features of the study protocol.

The sample characteristics were described using medians and interquartile ranges (IQR). Categorical data were presented as absolute values (n) and relative frequencies (%). Statistical analyses were performed using Stata Data Analysis and Statistical Software (STATA) version 16.0 (Stata Corp., College Station, TX, USA). Alpha levels < 0.05 were considered significant.

Qualitative data from focus group discussions were audio-recorded and transcribed verbatim. Data were coded into themes and classified as: favourable factors and health events and not favourable using the content framework developed prior to the data collection. Quotes from participants were used to illustrate the themes.

## 5. Results

The study participation flowchart is presented in Figure 1. Detailed information about recruitment strategy and reasons for non-eligibility and non-participation of eligible women has been presented elsewhere [14]. Briefly, 767 women were invited to take part and 92 were initially deemed eligible. In total, 50 women declined the initial invitation, and 625 did not meet the inclusion criteria. Although the eligibility rate (92 out of 767 referred or 12%) was below the threshold of 15%, only 6.5% of women (50 out of 767 referred) were not interested in taking part in the study.

All participants had the physical and mental capacity to consent to participation at the beginning of the study. After consent, the research team applied a preliminary eligibility checklist and those eligible undertook a full-mouth oral examination to confirm the presence of periodontitis. The main reason for exclusion was advanced gestational age (>20 weeks) at the first prenatal appointment (n = 318) followed by the presence of caries (n = 64) and the absence of periodontitis (n = 58). Baseline data were collected immediately after confirmation of periodontitis and prior to randomisation. In total, 8.7% of eligible participants (8 out of 92 eligible women) declined consent before randomisation. During the trial, two additional women withdrew consent after the randomisation (Figure 1).

Feasibility findings regarding all quantitative indicators from all data sources are outlined in the adapted checklist for feasibility studies [16] and displayed in Table 1. The recruitment rate of 2 women/week (70 randomised women during 32 weeks of recruitment) was above the target of 1.7 women/week. The retention rate at the first follow-up wave (T1) was 89.8% (62 of 69 randomised participants). The retention rate at the end of the study (T2) was 79.7% and met the ”acceptance” threshold of ≥70.0%.

All randomised participants accepted their group allocation (early vs. late PT and fortified vs. plain milk) and there were no instances of reported un-blinding (fortified or placebo sachet) to the research team. There were no significant differences among groups regarding the main socio-demographic, nutritional status, and clinical parameters of periodontitis at baseline (Table 2).

Completers (n = 55) did not significantly differ from those who dropped out (n = 15) regarding sociodemographic characteristics at baseline (Table 3).

Data collection of the majority of outcome measures was considered feasible by the field workers, and there was no report by participants of any major complaints on feeling burdened or frustrated with data collection or taking part in the study. However, it was reported that on some occassions, women were late for the appointments or missed scheduled appointments due to lack of childcare, other commitments, and violence in the local area. The team also reported that women were more likely to re-book appointments after childbirth. In the end-of-project evaluation, fieldworkers reported that the questionnaire was extensive, but the use of electronic questionnaires facilitated the data collection process.

The average adherence to the treatment, measured by the number of visits divided by the total number of PT recommended, was 98.8% and 29% in the early and late treatment groups, respectively. The adherence to early PT was above the threshold of acceptance (70%). Adherence to late PT was low. However, the late PT is not considered an active intervention arm. It was an alternative to a control group without PT.

The average adherence to milk consumption, measured by the number of sachets consumed divided by the total number of sachets provided to the pregnant women, was 82.4% in the fortified group and 88.1% in the placebo group, respectively. Both values were above the threshold of 80%. No severe adverse events were recorded suggesting that the intervention was well tolerated.

To facilitate the understanding of the results related to the acceptability of the milk, the responses of a five-point Likert-scale question were merged into three categories: dislike, neither like nor dislike (considered as indifferent), and like. The fortification group reported significantly higher acceptability compared to the placebo group in the second (*p* = 0.034) and third trimesters (*p* < 0.001). In the second trimester of gestation (T0), 74% of the fortified group and 59% of the placebo group liked the milk (Figure 2). In the third trimester of gestation (T1), 84% of the fortified group and 56% of the placebo group liked the milk.

### Qualitative Findings

The qualitative data are presented in Table 4 including quotes from participants to illustrate the key emergent themes.

Regarding milk consumption, some women complained about the diet being monotonous and disliked the taste of pure milk. To circumvent this potential acceptability issue, the team provided women with a list of recipes they could use to prepare meals/drinks with the milk powder (e.g., smoothies and porridges). Only a few women reported difficulties with the mode of preparation, and the majority reported good culinary skills. Women also appreciated the provision of a bottle shaker and mentioned that it facilitated mixing the milk, the content of the sachet, and water. A social media channel was used to promote interactions among participants, and women used this channel to share new recipes.

Consumption of milk and vitamin or mineral supplements during pregnancy was considered favourable among participants. No women reported milk allergy or lactose intolerance during the study or raised concerns regarding the safety of milk, vitamin D, and calcium consumption. However, some women reported nausea when consuming the milk at the beginning or towards the end of the pregnancy.

Provision of additional milk to family members (e.g., young children) regardless of group allocation was favourably seen by participants as it prevented them from sharing their milk provision with the rest of the family and consequently interfering with the adherence to milk consumption. Women also viewed positively the opportunity to receive dental treatment during pregnancy or after childbirth. However, pain and discomfort were reported during the PT, although women rated the pain as bearable. Women reported receiving valuable information on how to use dental floss correctly and were satisfied with the treatment results.

Some women reported difficulties in attending the monthly visit to collect the milk due to issues with childcare and lack of money to pay for transportation. Additionally, women felt unsafe when going to the health centre. Women were allowed to bring their children to the health centre, but the study did not provide childcare facilities. Travel expenses were refunded, and women were given a small compensation for attending the face-to-face visit, but money was not given up-front. To mitigate issues related to attendance, milk was delivered at home and women were contacted via phone calls to gather information regarding milk consumption, presence of adverse effects, and any potential barrier to compliance with the study protocol. Women reported varied levels of support. Some could count on a family network and some showed strong links with their mothers. However, others did not have a social network of support.

## 6. Discussion

Overall, in this evaluation using different qualitative and quantitative data sources, we found that undertaking an RCT of this intervention was feasible and that the intervention itself was considered safe, acceptable, and well tolerated by participants. However, minor modifications will be necessary for the full-scale trial.

Although the recruitment rate (2 women/week) exceeded the target, the overall recruitment goal of 120 women was ambitious and not achieved. This was due to the inclusion of only one study site instead of two and several recruitment interruptions due to general strikes, public manifestations due to the political and economic instability, public holidays, floods, and an episode of armed robbery. A full description of recruitment challenges is presented elsewhere [14]. Although only few women were not interested in taking part in the study (declined invitation), the recruitment rate could have been enhanced by closer engagement with medical doctors, who could promote the relevance of the study to patients.

Eligibility rate was below the threshold of acceptance (12% vs. 15%), mainly due to women starting prenatal care at advanced gestational age and the presence of open cavities. The health care centre closure for 16 weeks due to a general strike was an unpredictable factor that significantly delayed the onset of prenatal care. Initially, only women during the 1st trimester were considered eligible but this was revised to include women up to 20 weeks’ gestation. Our study population was young, and we anticipated that applying a strict diagnostic criterion for chronic periodontitis, which tends to develop with age, would have resulted in a lower eligibility rate. Therefore, periodontitis was defined as the presence of ≥1 tooth with at least one periodontal site with ≥4 mm of clinical attachment loss with the presence of bleeding on probing on the same site. The presence of bleeding on the same site ensured the existence of local inflammation. However, if we had applied a strict criterion (e.g., ≥2 teeth with at least one site with ≥4 mm of clinical attachment loss), only one woman would have been excluded. To further enhance the eligibility rate for the large-scale trial, the research team will need to provide dental treatment before randomisation; thus, women with open cavities, representing 10.2% of exclusions, could be potentially included in the study. Other exclusion factors were related to patient safety, and they should remain in the full-trial.

Dental caries and periodontitis are the most common oral health diseases in the adult population [18]; the prevalence of periodontitis in this study was 42%. This was a conservative estimate because it did not consider women with caries who were excluded after the full dental screen and not further assessed. A proportion of those women might have also presented with periodontitis as bacterial plaque is the main etiological factor for periodontitis. Although earlier literature points to a causal relationship between *Streptococcus mutans* and the development of caries, contemporary research on microbiome shows that both caries and periodontitis are not caused by singular pathogens and they seem to result from a perturbation among relatively minor constituents in local microbial communities leading to dysbiosis [19]. In this scenario, the presence of caries might increase the risk of periodontitis or vice-versa [20,21].

The reach and potential generalisability of results are relevant aspects when designing interventions. Our findings show that women had 12 years of education and a monthly per-capita income of USD 126.7. This indicated that we might have excluded low educated and very low-income women even though the study site was located in a deprived area in Duque de Caxias. Women with poor oral health (extensive caries and few natural teeth) were not eligible and those women are more likely to be of low socioeconomic status. Social determinants such as low income and limited access to health systems are correlated with the development of periodontal diseases. According to Vettore et al. [22], income inequality was associated with severe periodontal disease in population-based research in Brazil. The authors showed that the risk of moderate to severe and severe periodontal disease was higher among Brazilian adults with low dental health care coverage. Offering dental treatment to those without severe cavities before randomisation might help enhance the external validity of the trial.

The randomisation created balanced groups even in a small sample size, demonstrating that the randomisation strategy was adequate. Furthermore, women did not express concerns about being allocated to the late treatment or the placebo group. As informed by the findings generated during the patient consultations prior to study implementation, all women were given additional milk to share with their family and received PT either during pregnancy or after childbirth. This design ensured that all participants benefitted from the intervention and might have improved acceptability, retention, and satisfaction with the study design instead of having a control group not receiving periodontal treatment. The delayed PT was offered at 6–8 weeks postpartum to all control women. However, adherence to PT after childbirth was low. This arm was included to offer participants the opportunity of having PT even though they were allocated in the control group. The observed low adherence to late PT is in line with the literature indicating that caring for their family is a high priority for women with young children and they might sacrifice their own health care needs [23,24].

There were some interesting aspects of the dietary intervention that emerged in this mixed-methods evaluation. The acceptability of the dietary intervention was below the target of 90%, particularly among the placebo group in both T0 and T1. However, 30% of the placebo group also expressed that they did not like nor disliked the milk. The qualitative data showed that women found it monotonous to consume pure milk twice a day. Some women who reported nausea found it difficult to consume 200 mL of milk twice daily. This might have influenced the women’s responses in the Likert-scale questionnaire and explained the reason why the acceptability indicator was low. However, adherence to the dietary intervention was high in both groups. Although women were advised to consume pure milk or to use the milk in preparations including porridge and smoothies, these preparations did not affect the bioavailability of calcium and vitamin D. However, some women reported mixing the milk with chocolate powder, which contains oxalate affecting calcium absorption. For the large-trial, women will receive a brochure with all recipes gathered during the feasibility trial (including porridges, smoothies, and puddings) to make the preparation more appealing and less monotonous to participants without compromising the bioavailability of calcium and vitamin D. Additionally, the intervention will be slightly modified to provide only one dose/serving of milk (with and without fortification) instead of two servings daily. However, the total amount of calcium and vitamin D offered will be maintained.

This multi-component intervention had no observable adverse effect. Given the small sample size, it is not possible to claim that the intervention is completely safe, but adverse events in the large-scale trial are not likely to occur. Additionally, there was no report of any major issues regarding participation burden, but the research team noticed that women with caring responsibilities preferred home-delivery of the milk to the monthly collection at the health centre despite the financial compensation for their time. The team also reported that women were more likely to re-book study visits after childbirth. To enhance data completeness, the final assessment should coincide with the date of the routine maternal or newborn care.

Retention rate at the second follow-up was within the target and completers and drop-outs did not significantly differ in regard to sociodemographic characteristics at baseline. The overall similarity between completers and dropouts suggests that the presence of dropouts in this feasibility clinical trial does not substantially influence the generalisability of results obtained solely from study completers. Since this is a clinical trial with specific inclusion and exclusion criteria, sampling bias cannot be ruled out. Furthermore, our sample comprised women who were eligible after dental screening and contributed data up to 6–8 weeks after childbirth, hence a homogeneous sample of the initial eligible participants.

The field of nutrition interventions for dental diseases, particularly periodontitis, during pregnancy is newly emerging. This is the first 2 × 2 randomised controlled feasibility trial of milk fortification and PT tailored for delivery in a low-income setting. Vitamin D is a potent immunomodulator due to its anti-inflammatory effect through the inhibition of cytokine production by immune cells [25]. Vitamin D may, therefore, be beneficial in the treatment of periodontal disease, in which host-defense cells activated by the bacterial release of inflammatory mediators destroy supporting periodontal tissues, including connective tissue and alveolar bone [26]. Calcium and vitamin D have been hypothesised to act jointly rather than independently. Previous reports have shown that joint supplementation is much more efficient in influencing metabolic profiles than single calcium or vitamin D supplementation [27]. Additionally, the literature points out that intake of calcium within dietary recommendations is associated with a lower risk of periodontitis and tooth loss only among those with a higher intake of vitamin D [12,28].

This feasibility clinical trial makes a valuable contribution to the design of promising coadjutant non-pharmacological or non-invasive interventions for periodontitis. Practical implications drawn from this study can be applied to other studies in similar settings. However, interpretation of the findings should be in light of study limitations. Our findings from acceptability questionnaire responses and participants’ views and experiences in the focus group discussions and phone calls might have been influenced by social desirability bias when participants tend to answer in a way they perceive to be socially acceptable or expected. Additionally, we were not able to collect information on the reasons for dropouts and missing appointments for all women.

In conclusion, the study design was deemed feasible, and the intervention was acceptable and safe. A full-scale trial is now warranted to establish clinical effectiveness, as this multi-component intervention might help women to deal with issues related to metabolic disorders and inflammation associated with periodontitis, which may have important health consequences for the pregnant woman and her offspring. Moreover, we consider that this intervention can potentially achieve wide application in low-risk prenatal care programmes in deprived areas in Brazil.

## Figures and Tables

**Figure 1 ijerph-17-08023-f001:**
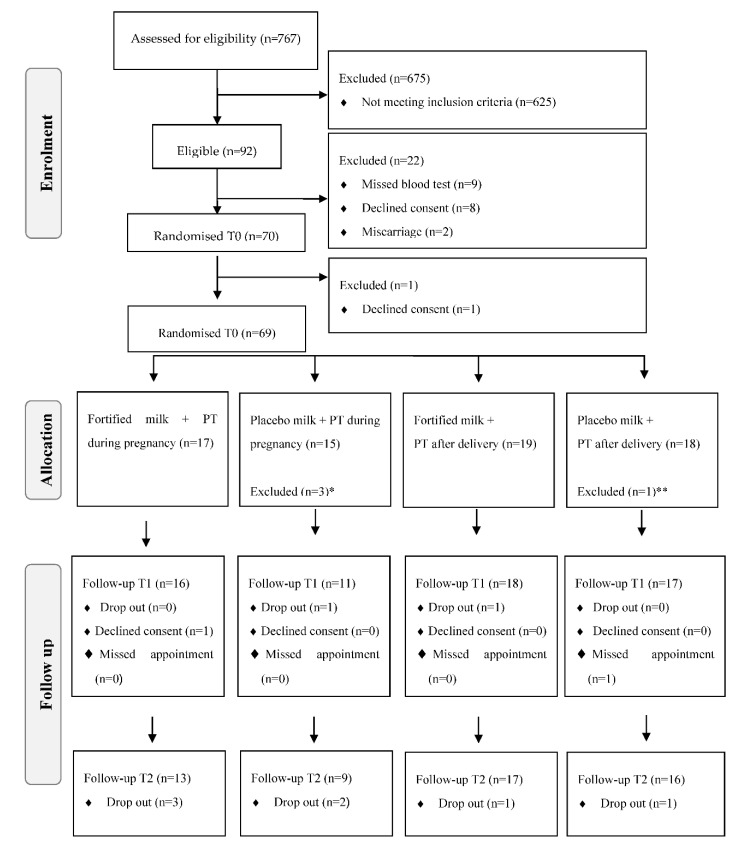
Flowchart of enrolment, allocation, and follow up of the pregnant women from a low socioeconomic area in Rio de Janeiro. PT: Periodontal therapy. * Not started periodontal treatment (n = 1) or milk consumption (n = 1) or both (n = 1). ** Not started milk consumption (n = 1).

**Figure 2 ijerph-17-08023-f002:**
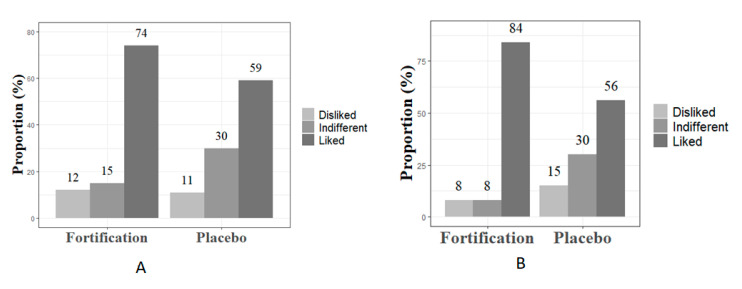
Milk acceptability between pregnant women with periodontitis from a low socioeconomic area in Rio de Janeiro, Brazil. Figure (**A**) = T0 and (**B**) = T1.

**Table 1 ijerph-17-08023-t001:** Adapted checklist for feasibility, acceptability, tolerability, and adherence of trial design, study procedures and intervention.

Indicators	Threshold	Data Source	Descriptive Outcome	Decision
Eligibility rate ^a^ N of eligible participants/total n of participants referred to the study	**A:** ≥15% **M:** 15–10% **R:** <10%	Recruitment screening log	767 women were referred and 92 were considered eligible after dental screening. Eligibility rate was 12%	Modify
Participation rate ^a^ N of randomised participants/n of eligible participants after dental screening	**A:** ≥75% **M:** 74–70% **R:** <70%	Recruitment screening log	92 women were eligible after dental screening and 70 were randomised. Participation rate was 76.1%	Accept
Recruitment rate ^a^ N of randomised participants/total n of recruitment weeks	**A:** ≥1.7 women/week **M:** 1–1.6 women/week **R:** <1 women/week	Recruitment screening and participant flow logs	Actual recruitment of 2 women/week (70 randomised women in 32 weeks)	Accept
Retention rate N of randomised participants remaining in the study/total n of randomised participants	**A:** ≥80% **M:** 79–70% **R:** <70%	Participant flow log	In total, 70 women were randomised. 69 women remained at the baseline, 62 in the 2nd follow-up and 55 in the 3rd follow-up Retention rate: 78.6%	Modify
Adherence to milk consumption N sachets consumed/total n of sachets provided to participants	**A:** ≥80% **M:** 79–60% **R:** <60%	Participant flow log	Fortification group: 82.4% Placebo group: 88.1% Overall:85.2%	Accept
Adherence to periodontal therapy N of therapy sessions completed per PT group/total n of therapy sessions offered per PT group	**A:** ≥70% **M:** 69–60% **R:** <60%	Participant flow log	Early PT group: 98.8%	Accept
Tolerability of intervention N of serious adverse events related to the intervention	**A:** no events **M:** tolerable for the majority of participants **R:** any serious adverse event related to the intervention	Routine phone calls Field notes Blood test results	No adverse event reported	Accept
Acceptability of random allocation N of randomised participants accepting recruitment allocation	**A:** ≥95% **M:** 94–90% **R:** < 90%	Participant flow log Field notes	All randomised participants accepted their allocated group (100% acceptability)	Accept
Acceptability of milk consumption ^b^ N of participants who liked the milk in T0 and T1/total n of participants who answered the questionnaire in T0 and T1	**A:** ≥90% **M:** 89–70% **R:** <70%	Study questionnaire	T0 74% in the fortification group liked the milk 59% in the placebo group liked the milk T1 84% in the fortification group liked the milk 56% in the placebo group liked the milk	Modify for the fortified group Reject for the placebo group
Balanced groups at baseline The ability of random sequence generation to produce comparable groups	**A:** no sig differences **M:** 1–2 sig differences **R:** >2 sig differences	Descriptive statistics	No significant differences in the main socio-demographic characteristics	Accept
Blinding N of un-blinding cases reported by the trial coordinator/total n of randomised participants	**A:** <10% **M:** 10%–15% **R:** ≥15%	Field notes	No un-blinding cases reported	Accept
Feasibility of data collection Reported ability of researchers of applying questionnaires and complete activities on the study protocol	**A:** no major reported difficulty **M:** few minor reported difficulties **R:** any major reported difficulty	Field notes End-of-study evaluation survey	No reports of difficulties	Accept
Tolerability of data collection and study participation N of complaints related to taking part in the study (visits to the centre, filling up questionnaires, blood tests, etc.)	**A:** no major complaint **M:** few minor complaints **R:** any major complaint	Field notes	No participant reported any major complaints on feeling burdened or frustrated with data collection or taking part in the study	Accept

**A**, acceptance; **M**, modification; **R**, rejection N and n, number (s); Sig, significant. PT, periodontal therapy ^a^ Full data reported elsewhere [14]. ^b^ Five-point Likert-scale question regarding acceptability of the milk powder.

**Table 2 ijerph-17-08023-t002:** Baseline characteristics of pregnant women from a low socioeconomic area in Rio de Janeiro, Brazil.

Variables ^a^	Total	Early PT (during Pregnancy)		Late PT (After Delivery)		*p* -Value ^d^
		Plain Milk	Fortified Milk	Plain Milk	Fortified Milk	
	Median (IQR)	Median (IQR)	Median (IQR)	Median (IQR)	Median (IQR)	
Age (year)	28.0 (7.0)	29.5 (6.0)	28.0 (9.0)	25 (10.0)	29.0 (7.0)	0.51
Gestational age (week)	15.0 (5.0)	14.5 (5.0)	16.0 (2.0)	13.0 (4.0)	16.0 (5.0)	0.17
Education (year)	12.0 (3.0)	12.0 (2.0)	12.0 (3.0)	11.0 (4.0)	11.0 (2.0)	0.96
Monthly per-capita ^b^ income (USD)	126.7 (94.9)	147.8 (93.3)	100.0 (69.1)	126.7 (207.5)	151.6 (131.6)	0.19
Pre-pregnancy BMI (kg/m^2^)	26.3 (9.5)	25.9 (8.3)	23.9 (8.5)	22.4 (12.8)	28.6 (7.7)	0.77
Pocket depth (mm) ^c^	4.2 (0.3)	4.2 (0.4)	4.3 (0.3)	4.2 (0.3)	4.2 (0.4)	0.50
Clinical attachment loss (mm) ^c^	4.2 (0.3)	4.3 (0.4)	4.3 (0.3)	4.2 (0.2)	4.2 (0.3)	0.81
Sites with bleeding on probing (%) ^c^	16.0 (21.0)	23.0 (31.0)	19.0 (11.0)	16.0 (17.0)	12.0 (14.0)	0.36
	**N (%)**					***p*** **-Value ^e^**
Marital status						
Living with partner	60 (87.0)	16 (88.9)	13 (76.5)	13 (86.7)	18 (94.7)	0.43
Other ^f^	9 (13.0)	2 (11.1)	4 (23.5)	2 (13.3)	1 (5.3)	
Self-reported skin colour						0.38
White	10 (14.5)	3 (16.7)	3 (17.6)	-	2 (10.5)	
Other	59 (85.5)	15 (83.3)	14 (82.3)	15 (100.0)	17 (89.5)	
Parity ^g^						0.86
0	24 (34.8)	5 (33.3)	7 (41.2)	5 (27.7)	7 (36.8)	
≥1	45 (65.2)	10 (66.6)	10 (58.8)	13 (72.2)	12 (63.1)	
Current smoker						0.82
No	61 (88.4)	16 (88.9)	16 (94.1)	13 (86.7)	16 (84.2)	
Yes	8 (11.6)	2 (11.1)	1 (5.9)	2 (13.3)	3 (15.8)	
Alcohol consumption						0.89
No	57 (82.6)	15 (83.3)	15 (88.2)	12 (80.0)	15 (78.9)	
Yes	12 (17.4)	3 (16.7)	2 (11.7)	3 (20.0)	4 (21.0)	

The baseline period was between gestational weeks 6 and 21. BMI, Body Mass Index. IQR, Interquartile range (the difference between upper and lower quartiles).^a^ n = 69; ^b^ value originally measured in Brazilian Reais (BRL) but converted to USA dollars (USD). Exchange rate in February 2019, BRL 3.75 = USD 1; ^c^ n = 67; ^d^ Kruskal–Wallis test; ^e^ Qui-squared test. ^f^ Other, not living with a partner, or do not have a partner. ^g^ Parity, number of parturitions.

**Table 3 ijerph-17-08023-t003:** Sociodemographic and maternal baseline characteristics comparisons of the pregnant women with periodontitis with complete data (three measures) and one or two measures from a low socioeconomic area in Rio de Janeiro, Brazil.

**Variables**	**Pregnant Women**		***p*** **-Value ^a^**
	**Complete (Three Measures)** **N = 55**	**One or Two Measures** **N = 15**	
	**Median (IQR)**	**Median (IQR)**	
Age (years)	29.0 (8.0)	25.0 (8.0)	0.115
Gestational age (weeks)	16.3 (4.7)	16.4 (3.1)	0.517
Schooling (years)	12.0 (2.0)	11.0 (4.0)	0.270
Monthly per-capita income (USD) ^b^	130.0 (104.9)	124.75 (140.0)	0.621
Prepregnancy BMI (kg/m^2^)	27.6 (9.5)	22.7 (8.4)	0.161
**Variables**	**n (%)**	**n (%)**	***p*** **-Value ^c^**
Marital status			0.077
Living with partner	49 (90.7)	11 (73.3)	
Other ^d^	5 (9.3)	4 (26.7)	
Self-reported skin colour			0.500
White	7 (13.0)	1 (6.7)	
Black or mixed	47 (87.0)	14 (93.3)	
Parity			0.894
0	19 (35.2)	5 (33.3)	
≥1	35 (64.8)	10 (66.7)	
Alcohol			
No	47 (87.0)	10 (66.7)	0.066
Yes	7 (13.0)	5 (33.3)	
Current smoker			
No	50 (91.0)	11 (78.6)	0.34
Yes	5 (9.0)	3 (21.4)	

BMI, Body Mass Index. IQR, Interquartile range (the difference between upper and lower quartiles). Analyses were performed among all randomised participants (n = 70). ^a^ Kruskal–Wallis test. ^b^ Value originally measured in Brazilian Reais (BRL) but converted to USA dollars (USD). Exchange rate in February 2019, BRL 3.75 = USD 1. ^c^ Qui-squared test. ^d^ Other, not living with a partner, or do not have a partner.

**Table 4 ijerph-17-08023-t004:** Quotes related to the content included in the matrix.

Category	Sub-Categories	Factors	Quotes
**Dietetics and culinary skills**	Mode of preparation and consumption New recipes Difficulties in conventional preparation	Favourable: Preparation was considered easy. Women reported consumption of smoothies (milk blended with fruits). Women had basic utensils at home for simple recipes (porridges and smoothies), but the provision of a shaker bottle helped with the preparation. Unfavourable: dislike of the test of pure milk; consumption of milk daily was considered monotonous.	*“In the beginning, it was very difficult to adapt to taking it twice and at the end of the pregnancy, I was already sick.* *Sometimes I took it pure but I got tired of it. Mixing with fruits, yogurt or in the porridge is much better.”* *“I add the milk powder, sachet and powdered cereal in the saker and carry it with me. I always have a bottle of water and the shaker is very handy.”*
	Milk intolerance Cultural belief	Favourable: milk was considered a healthy food. Unfavourable: Some women reported nausea when consuming milk and sachet.	*“In the beginning, it was very good because I was not eating, I was’ losing weight. So for me, milk was my only food.* *Milk is good for our health. Vitamins and milk are good for the baby.”* *“I was vomiting in the beginning. I could not take it. Now it is okay.”*
**Sharing of food with other family members**	Family access to food Eating together as a family Food distribution	Favourable: Provision of whole milk to the children prevented the sharing of the milk provided to women with their family. Unfavourable: Some women shared milk and sachets with their children. Sharing was sporadic and related to children’s curiosity.	*“I gave it to my son. He just wanted to taste it.”* *“The children like milk but I did not share my mine with them. They had their milk.”*
**Health care needs**	Dealing with pain Dental care experience	Favourable: Women reported a positive outcome after PT. Women considered dentists competent. Women trusted health care professionals. Unfavourable: Women complained about discomfort and pain during the PT.	*“It hurts but it very good (…) my teeth are now sparkling clean.”* *“I felt discomfort but it was bearable.”* *“I see amazing results.”* *“The dentist was excellent.” “Everybody there is nice. She explained how to use dental floss. I have never used it properly.”*
	Access to health care	Favourable: refund of transportation cost and home delivery of milk was appreciated by the women. Unfavourable: Lack of money; lack of safety and fear of violence; competing priorities prevented women to attend the visits.	*“I avoid going there too often because of the lack of security in the area.”* *“There are times when I do have money at home to go to the health centre.”* *“Home delivery was convenient. It is difficult to go out when you are busy and have kids at home.”*
**Social support network**	Family support Social challenges	Favourable: Some women had support from their mothers, the father of the baby or wider family. Unfavourable: Some women lacked support from their mothers, the father of the baby, or wider family	*“(…) I count on my mum to stay with my daughter when I go to the health centre.”* *“I have my mum. She helps me a lot. (…) She reminds me to take the milk.”* *“I do not have anybody to help with my kids. I leave them at school. I manage things on my own.”*

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
