# Peer review of "Vitamin D and Calcium Milk Fortification in Pregnant Women with Periodontitis: A Feasibility Trial"

_ijerph, 2020, doi:10.3390/ijerph17218023_

Round 1

Reviewer 1 Report

Thank you for the opportunity to contribute to the peer review process for the original study submission manuscript entitled “Acceptability, adherence and retention of a feasibility randomised trial on periodontal treatment and vitamin D/ Calcium milk fortification among pregnant women: a mixed-methods evaluation”. It is an interesting and relevant study where the authors discuss the possibility of a trial on milk fortification with calcium and vitamin D (Ca+VitD) and periodontal therapy (PT) among low income pregnant women with periodontitis. The manuscript has clear aims and attended them in a satisfactory manner.

Just minor revisions are required that are indicated below:

Line 137 – remove the period in the middle of the sentence

Line 143 – T0 and T1 should be better specified like: second trimester (from the 14th to 27th gestational week)…

Line 260 – Table 2 is not cited in the text

Tables 2 and 3 – please, review the Monthly per capta income. Both tables present similar values, but in Table 2 the label is in R$ and in Table 3 it is US$

Table 2. It would be much more informative if you report Q1-Q3 instead of IQR

Line 268 – “those women”

Line 276 – This sentence is part of Tables 3, it is confuse the way it is here. Just a matter of formatting,

Line 288 – “…low However, …” check the sentence

Line 351 – “by closer engagement with GPs,…” Please explain what a GPs means, the acronym did not appear before

Author Response

Reviewer 1

Point 1 Line 137 – remove the period in the middle of the sentence

Thanks for pointing this out. We removed the period. We also made corrections throughout the text and we hope we have identified all minor grammatical errors and typos and improved the overall readability of the manuscript.

Point 2 Line 143 – T0 and T1 should be better specified like: second trimester (from the 14th to 27thgestational week).

Thanks for this comment. We added information about the weeks of gestations when the data was actually collected (Lines 156157).

Point 3 Line 260 – Table 2 is not cited in the text

Thanks for spotting this mistake. We include Table 2 in the text (line 261). Additionally, we have ensured that all tables and figures are cross-referenced in the text.

Point 4 Tables 2 and 3 – please, review the Monthly per capta income. Both tables present similar values, but in Table 2 the label is in R$ and in Table 3 it is US$

Thanks again for spotting this mistake. We confirm that the data is presented in US$. We have corrected Table 2 to ensure that information is now consistent.

Point 5. Table 2. It would be much more informative if you report Q1-Q3 instead of IQR

IQR is the difference between 75th and 25th percentiles, or between upper and lower quartiles, IQR = Q3 − Q1. To aid interpretation of results we spelt out the definition of IQR in the text. I hope the reviewer finds this satisfactory. 

Point 6 Line 268 – “those women”

We appreciate this comment. We corrected the sentence. It now reads “Completers (n=55) did not significantly differ from those who dropped out (n=15) regarding sociodemographic characteristics at baseline (Table 3)”.

Point 7 Line 276 – This sentence is part of Tables 3, it is confuse the way it is here. Just a matter of formatting,

We appreciate that this might be confusing for the readers. We arranged the text and hope the changes have improved the text flow.

Point 8 Line 288 – “…low However, …” check the sentence

Thanks for pointing this out. A period was missing right after the word “low”. The sentence was corrected.

Point 9 Line 351 – “by closer engagement with GPs,…” Please explain what a GPs means, the acronym did not appear before

Thanks again for the pertinent comment. GP (general practitioner) is a well-known term in the UK. However, we appreciate that this is not a common term in other countries. We have replaced the term by the word “medical doctor” as it is widely understood by different readers.  

Reviewer 2 Report

The manuscript is correctly written, it is coherent between objectives, results and conclusions. It analyzes the feasibility and adherence of a clinical trial. The methodology is written in detail and the results are adequately shown, specifically those related to qualitative analysis.
And it shows the feasibility of applying the clinical trial to large populations.

Although the authors refer the reader to previously published articles where the techniques employed are described in detail, it might be desirable to expand the methodology with respect to the methodology for establishing the periodontal status of the women included. Were the staff who collected the periodontal index calibrated? Who calibrated them? It would also be desirable to include in the discussion how the authors will establish the possible impact of Calcium and Vitamin D on periodontal disease.

Author Response

REVIEWER 2

The manuscript is correctly written, it is coherent between objectives, results and conclusions. It analyzes the feasibility and adherence of a clinical trial. The methodology is written in detail and the results are adequately shown, specifically those related to qualitative analysis. And it shows the feasibility of applying the clinical trial to large populations.

Thanks for the positive comment

Point 1. Although the authors refer the reader to previously published articles where the techniques employed are described in detail, it might be desirable to expand the methodology with respect to the methodology for establishing the periodontal status of the women included.

Thanks for this comment. We agree with the reviewer that readers would benefit from more details about the oral examination (lines 130-134). We have provided more information about the procedures. We hope that the reviewer finds this change satisfactory.

Point 2. Were the staff who collected the periodontal index calibrated? Who calibrated them?

Thanks for this relevant comment. The oral examination was performed by a trained dentist who was calibrated by the research dentist. This information is now included in the text.

Point 3. It would also be desirable to include in the discussion how the authors will establish the possible impact of Calcium and Vitamin D on periodontal disease.

Thanks for this interesting point. This paper focus on feasibility indicators and the main objective of this study is to report the acceptability, adherence, attrition rate, and tolerability of data collection and study participation by taking a mixed-methods approach. We have a wealth of qualitative and quantitative data on feasibility indicators/ parameters. Therefore, the discussion was focused on a critical reflection on these indicators. This feasibility study was not designed and powered to find a statically significant effect. All analyses were exploratory, and results will be used to inform the sample size estimation for a large-scale trial. However, we have included a brief description of the potential impact of calcium and vitamin D on periodontitis in the discussion (Lines 458-468).

Reviewer 3 Report

  • Title is a little too much to digest, consider:“Periodontal treatment and vitamin D/ Calcium milk fortification among pregnant women: a feasibility randomized trial”
  • Line 93 & 95: what do you “low risk”, risk of what?
  • Line 106-7: Please describe who performed the dental screening, where, and how?
  • Line 61: Perhaps you want to give a more spherical view on the premise of an effect for Vitamin D on periodontitis, that there are reports supporting some effects and others which do not.
  • Another consideration may be the questioning of the existence of the above effects on gingivitis in these women.
  • Your periodontal parameters for inclusion are rather lenient (more than 1 pocket deeper than 4mm), it is my feeling that this may not be representative or truly reflect the feasibility of a future trial. It is my knowledge that severe periodontitis may benefit more from Vit D supplementation. If you look for moderate/severe periodontitis there will be an impact on your recruitment rate for sure.
  • If you keep this criterion you still need to exclude patients who may have pockets due to other reasons. For example, gingival enlargement may be common in pregnant women.
  • Line 313: Is the way of using the milk powder (drink/smoothie/cream) affecting bioavailability of VitD/calcium Is it possible that you add additional noise to the effects that you plan to record in the future RCT? Can this be validated?

Author Response

REVIEWER 3

Point 1. Title is a little too much to digest, consider: “Periodontal treatment and vitamin D/ Calcium milk fortification among pregnant women: a feasibility randomized trial”

We appreciate the reviewer’s comments and the suggestion to shorten the title. Thus, we changed the manuscript title to “Feasibility Indicators of a Periodontal treatment and vitamin D/ Calcium milk fortification trial among pregnant women”

Point 2. Line 93 & 95: what do you “low risk”, risk of what?

Thanks for your comment. Low risk refers to low-risk pregnancy. We agree that the text is confusing. We revised the sentence, and we hope that the text is now clear (Lines 94-96).

Point 3 Line 106-7: Please describe who performed the dental screening, where, and how?

Thanks for this pertinent comment. The dental screening was conducted by a trained dentist. One dentist performed the screening and the periodontal treatment, and a second blinded dentist performed the periodontal assessment.   We agree that readers would benefit from more details about oral screening. We included more information about the oral examination in the text (Lines 105-108).   

Point 4. Line 61: Perhaps you want to give a more spherical view on the premise of an effect for Vitamin D on periodontitis, that there are reports supporting some effects and others which do not.

Thanks for this comment. We agree that there is a certain level of inconsistency in the literature hence the need for a large and well-designed trial addressing this issue. We added brief information on the effects of vitamin D and calcium on periodontitis in the discussion (Lines 458-467). However, as this feasibility study was not designed and powered to investigate the effect of the interventions on periodontitis, but rather to inform an adequately powered large-scale trial, we prefer not weighing too much on a discussion we don’t have yet evidence to support.

Point 5. Another consideration may be the questioning of the existence of the above effects on gingivitis in these women.

Thanks for this comment. The literature shows positive effects on gingivitis, which are consistent with the effects seen on periodontitis. However, few studies have been conducted among pregnant women. Dietrich et. al. (2004) using data from the National Health and Nutrition Examination Survey concluded that lower levels of Vitamin D were associated with inflammation in the gums which is a precursor to gingivitis. An RCT found a dose-response relationship between vitamin D supplementation and anti-inflammatory effect among patients with gingivitis (Hiremath et al., 2013). A cross-sectional study evaluated the levels of vitamin D and calcium in serum of periodontally healthy, chronic gingivitis and chronic periodontitis patients with and without T2DM and consistently found that vitamin D and calcium levels are inversely correlated with random blood sugar and glycated haemoglobin and also probing pocket depth and clinical attachment loss, thus contributing towards an increase in periodontal disease severity. However, as stated above, we believe this discussion is beyond the scope of our study.

Point 6. Your periodontal parameters for inclusion are rather lenient (more than 1 pocket deeper than 4mm), it is my feeling that this may not be representative or truly reflect the feasibility of a future trial. It is my knowledge that severe periodontitis may benefit more from Vit D supplementation. If you look for moderate/severe periodontitis there will be an impact on your recruitment rate for sure.

This is a pertinent comment. Our study population is young and we anticipated that applying a more strict definition of chronic periodontitis, which tends to develop with age, would have resulted in a low recruitment rate. Therefore, periodontitis was defined as the presence of one or more teeth with at least one of the periodontal sites with ≥ 4 mm of clinical attachment loss with the presence of bleeding on probing on the same site. The presence of bleeding on probing (BOP) ensured the existence of local inflammation. However, applying a more strict definition (≥ 2 teeth with ≥ 4 mm) we would have excluded only one woman from our study.

Periodontal disease is unpredictable and slowly progressive so that a clinically relevant difference in pocket depth and attachment loss would not be expected or detectable during the intervention period. However, the absence of BOP is a patient observed health outcome and a reliable indicator of periods of periodontal disease stability and inflammation status. We included a comment about this issue in the discussion (Lines 377-384)

Point 7. If you keep this criterion you still need to exclude patients who may have pockets due to other reasons. For example, gingival enlargement may be common in pregnant women.

Thanks for your comment. The diagnosis criteria were based on the presence of ≥ 4 mm of clinical attachment loss and BOP on the same site. Therefore, gingival hyperplasia in addition to bleeding would also characterise local inflammation. We have included a comment about the diagnostic criteria in the discussion (Lines 377-382). We hope the reviewer finds our changes satisfactory.

Point 8. Line 313: Is the way of using the milk powder (drink/smoothie/cream) affecting bioavailability of VitD/calcium Is it possible that you add additional noise to the effects that you plan to record in the future RCT? Can this be validated?

Thanks for this relevant comment.  Women were asked to consume the milk during breakfast or afternoon snack to avoid concomitant intake of prenatal iron supplements routinely prescribed for consumption during main meals (lunch/dinner). All participants also consumed 60 mg/day ferrous sulphate from the second trimester, provided during standard prenatal care in Brazil. We included this information in the Methods section (Lines 124-128). 

Women were also advised to consume pure milk or to use the milk in preparations including porridge and smoothies (mixed with fruits). These preparations do not affect the bioavailability of calcium and vitamin D. However, some women reported mixing the milk with chocolate powder. Cocoa beans contain oxalate which affects calcium absorption. We included this information in the discussion (Lines 430 – 438).

Round 2

Reviewer 3 Report

Cogratulations to the Authors for addressing all the comments!

I still find the title hard to digest, nowadays less convoluted titles are advisable. 

I would still recommend a variation of the following:

"Vitamin D & Calcium diet fortification in pregnant periodontal patients:

a feasibility trial"

I see no point in mentioning periodontal treatment by the way.

Author Response

Thanks for the suggestion. We have amended the title to

'Vitamin D and calcium milk fortification in pregnant women with periodontitis: a feasibility trial.'